# MARSBot: A Bristle-Bot Microrobot with Augmented Reality Steering Control for Wireless Structural Health Monitoring

**DOI:** 10.3390/mi15020202

**Published:** 2024-01-29

**Authors:** Alireza Fath, Yi Liu, Tian Xia, Dryver Huston

**Affiliations:** 1Department of Mechanical Engineering, University of Vermont, Burlington, VT 05405, USA; alireza.fath@uvm.edu (A.F.); yi.liu.2@uvm.edu (Y.L.); 2Department of Electrical and Biomedical Engineering, University of Vermont, Burlington, VT 05405, USA; txia@uvm.edu

**Keywords:** microrobot, augmented reality, dynamics, structural health monitoring, human–robot collaboration, bristle-bots, additive manufacturing, augmented reality haptic interface, MARSBot, wireless microrobot

## Abstract

Microrobots are effective for monitoring infrastructure in narrow spaces. However, they have limited computing power, and most of them are not wireless and stable enough for accessing infrastructure in difficult-to-reach areas. In this paper, we describe the fabrication of a microrobot with bristle-bot locomotion using a novel centrifugal yaw-steering control scheme. The microrobot operates in a network consisting of an augmented reality headset and an access point to monitor infrastructures using augmented reality (AR) haptic controllers for human–robot collaboration. For the development of the microrobot, the dynamics of bristle-bots in several conditions were studied, and multiple additive manufacturing processes were investigated to develop the most suitable prototype for structural health monitoring. Using the proposed network, visual data are sent in real time to a hub connected to an AR headset upon request, which can be utilized by the operator to monitor and make decisions in the field. This allows the operators wearing an AR headset to inspect the exterior of a structure with their eyes, while controlling the surveying robot to monitor the interior side of the structure.

## 1. Introduction

The significance of infrastructures originates from the dependencies of societies on them, and as they are susceptible to operational stresses and environmental conditions, structural health monitoring (SHM) assists in increasing their serviceability [1]. In recent years, SHM has been incorporated to monitor the progression of damage-sensitive features [2]. The continuous inspection of structures can assist in detecting defects at early stages to prevent catastrophes [3].

Tian et al. in 2022 [4] reviewed the application of rigid robotics and mobile, multimodal, and soft robots in the SHM of bridges, providing information for choosing the proper system for the task. For concrete inspection, Huston et al. [5] developed a six-legged robot using dual-durometer pneumatic suction for climbing vertical walls. 

Vibrationally driven legged robots, called bristle-bots, follow complex mechanisms for movement, despite their simplicity in structure. In 2015, Cicconofri and DeSimone [6] used two degrees of freedom (DOFs) including horizontal movement and deviation of the angular displacement from the rest angle to model the movements of bristle-bots analytically. Becker et al. [7,8] used Lagrange equations of motion to model a two-DOF cylindrical chassis and developed prototypes, one with different inclination angles, and another with a camera for pipe inspection. Moreover, similar tube-like bristle-bots were developed for wired controlled steering in networks of pipes [9].

Further analysis of bristle-bots revealed that the direction of motion of similar systems can be inverted for forward and backward movements depending on the frequency of vibration applied to the system. Equivalent systems with an inactive robot on a vibrating base [10] and wired piezoelectric actuators mounted on 3D-printed millimeter-scale bristle-bots [11] were used to experimentally demonstrate this phenomenon.

In 2022, Fath et al. provided an overview of the mechanism of movement, power requirement, sensing, and control strategies for microrobots using piezoelectric materials to present a variety of innovations for solving the challenges of their autonomy and feasibility for running in confined spaces [12]. For pipe inspection, Brunete et al. [13,14] developed multi-configurable modular microrobots that adapt coordinately through a variety of pipes. Furthermore, microrobots with unique movement mechanisms, namely, adaptable wheeled microrobots for pipeline inspection, were fabricated and tested in 50 mm pipes using a wired connection to a DC motor [15].

Additionally, advancement in additive manufacturing led to the development of microrobots using a variety of driving and control technologies. In 2020, Hao et al. developed wired asymmetrical bristle-bots with a piezoelectric actuator that controls the steering by adjusting the frequency of vibrations [16]. Global control frequency and amplitude of substrate with an electrodynamic shaker were utilized to control the swarm of micro bristle-bots by collision-induced aggregations [17]. Additionally, the electric field can be utilized to control the shape of modular hydrogel microrobots to perform multiple tasks such as object delivery, grasping, and propulsion [18]. Special driving forces can be used, such as electrohydrodynamic thrust on microrobotic ionocrafts used for flying and PID control methods similar to the strategies used in quadcopters [19], as well as ionic conducting polymer film (ICPF) for actuating fish-like microrobots and controlling their speed by changing the frequency and amplitude of the input voltage [20].

Furthermore, magnetic fields are one of the primary driving sources for microrobots. Xu et al. [21] developed a group of magnetically driven millirobots with different magnetization directions, which allowed the decoupled independent control of the millirobots. In 2022, to overcome the challenges of movement in complex environments, multimodal needle-like microrobots were developed that consist of ferromagnetic nanoparticles. In this system, the external dynamic magnetic field is used as an actuator for a closed-loop control through visual feedback for path following [22]. Furthermore, advanced adaptive controllers have been proposed for the navigation control of magnetic helical microrobots in uncertain dynamic environments [23].

As additive manufacturing is straightforward and reliable, it is widely used in fabricating microrobots. For instance, MechaCell [24], a modular mechatronic device moving by rotation of unbalanced mass, was developed using 3D-printed parts. Moreover, a method for 3D printing millimeter-scale soft microrobots driven magnetically was investigated by using UV-cured auxiliary lines [25].

For collecting structural data in remote hard-to-reach areas, a framework consisting of an access point, sensor nodes, teams of small robots, and AR headsets was developed, which can enhance the human perception of structural health conditions [26,27]. Interactive robot control using AR is suggested to replace the traditional methods due to its practicality and the fact that it is more intuitive to use. This method was employed in controlling the Yaskawa Motoman SIA5D in automatic and manual mode [28]. In 2020, Makhataeva and Varol [29] reviewed the use of AR in four categories, namely, human–robot interaction, medical robotics, motion planning and control, and multi-agent systems. In addition to the use of AR, asynchronous P300-based systems [30] and eye blink electroencephalograms (EEGs) [31] are used as brain–computer interfaces for human–robot collaboration (HRC).

Despite the efforts to develop microrobots with a variety of driving sources and mechanisms, they either are mostly tethered or demand a high external voltage to operate. Developing such systems requires expensive lab equipment with high fabrication costs. Therefore, the field of SHM lacks small-scale robotic systems that are robust enough to move in remote confined spaces for inspection and require low-cost materials and parts for fabrication using affordable facilities.

The current research study aims to propose a method employing a commonly used low-cost microcontroller with Wi-Fi capabilities mounted on 3D-printed legs to perform SHM inspections in hard-to-reach areas. This system also benefits from the intuitive use of AR headsets that are simply an addition to the human visual inspection.

This paper introduces a series of dynamic cases to model the movement of microrobots using bristle-bot locomotion and additive manufacturing methods such as FDM and LCD 3D printing utilized to develop microrobots. Furthermore, the interfaces with the microrobots here presented benefit from using augmented reality headsets to steer the microrobot and see the first-person view of the microrobot while inspecting the structures, which is essential in structural health monitoring. Finally, the most practical prototype, which is a novel micro augmented reality steering bot (MARSBot) was tested for AR steering on wooden surfaces and structural health monitoring by inspecting Unistrut channels.

## 2. Materials and Methods

In this section, the mechanism of the microrobot movement including centrifugal yaw-steering control is explained, followed by a dynamic modeling of the system. Considering the motion of the microrobot, additive manufacturing processes were suggested and employed to develop multiple prototypes for comparison. Finally, two interfaces were programmed, and the steering of the microrobot was tested using HRC.

### 2.1. Dynamics

The microrobot used was a bristle-bot that moved with an eccentric mass. In this study, the dynamics problem of the system was simplified into six cases to derive equations and elaborate the concept for a better comprehension by the reader. This step is crucial to achieving the best design and development process. The overall process of the bristle-bot movement mechanism depending on the location of the eccentric mass is as follows.

In the first setup, shown in Figure 1a, the resting bristle-bot is in the horizontal position $x_{0}$ and vertical position $y_{0}$, while springs at the hip joints apply torque to maintain its height. In Figure 1b, the centered force $F_{CU}$ lifts the bristle-bot body while both legs rotate counterclockwise (CCW) and slide forward due to the spring force and a reduced friction force. Eventually, in Figure 1c, the centered force $F_{CD}$ pushes the bristle-bot down, and the normal force on the feet increases, to prevent them from slipping. Then, the legs rotate clockwise (CW), and the bristle-bot’s body moves forward.

In the second setup, shown in Figure 2a, a resting bristle-bot similar to that in the previous setup, was used. In Figure 2b, the bristle-bot rotates CCW and rises at the front due to the upward force $F_{FU}$. Following that, the front leg slides forward with reduced friction, while the rear leg does not move. Figure 2c displays the bristle-bot rotating CW to lower its front following the application of a downward force, $F_{FD}.$ In this step, the front leg does not move, due to the increased normal force and friction. However, the rear leg slides forward.

The dynamics of the microrobot were studied considering six cases, starting with a simple case with an eccentric mass on the surface. Then, more complex cases were examined to represent different aspects of the microrobot movement. The general assumptions made in the dynamic modeling were as follows:The curved legs of the microrobots were considered to be massless rigid links with torsional springs connected to the body.The body of the microrobot system was considered a rigid body.

The following cases were developed to derive the equation of motion based on Newton’s method [32].

#### 2.1.1. Case 1: Eccentric Mass on a Two-DOF System

This case considered a rigid body with an eccentric mass attached to its center of mass, including vertical and horizontal displacements on a surface with Coulomb friction force (Figure 3) [33].

The equation of motion based on (x, y) DOFs can be derived by decomposing the eccentric mass force $F_{m}$ applied to the system into horizontal $F_{x}$ and vertical $F_{y}$ components, as follows:(1)$$M\;\ddot{x}=F_{x}-f$$
(2)$$M\;\ddot{y}=N-Mg+F_{y}\;$$
where M is the mass of the whole system including the eccentric mass, f is the friction force, N is the reaction force, and g is the gravity; the details to derive each parameter are reported in Appendix B. In this case, if we assume that the system never leaves the ground:(3)$$N=Mg-F_{y}$$

Therefore, the system moves to the right if Equation (4) is satisfied
(4)$$F_{x}>f\;$$

In addition to the eccentric mass affecting the reaction force, which changes the friction force, placing the system on an anisotropic surface in which the coefficient of friction is higher in one direction than the other can lead the system to move in one direction [9].

#### 2.1.2. Case 2: Eccentric Mass on a Spring

Case 2 consisted of an eccentric mass attached to the system with one (y) DOF, mounted on a spring fixed to the surface (Figure 4).

The equation of motion can be derived for the one-DOF system as follows:(5)$$M\;\ddot{y}=F_{m}\sin\left(\omega t\right)+N-Mg$$
(6)N=−ky 
where k is the torsional spring stiffness.

#### 2.1.3. Case 3: Bristle-Bot with Eccentric Mass Mounted with an Offset

Considering the planar motions of a bristle-bot with eccentric mass and two legs, the system can be defined in (x, y) DOFs.

In this case, the system rested on torsional springs (Figure 5), which was equivalent to a system with curved legs. The equation of motion can be derived as follows [6]:(7)$$M\;\ddot{x}=-\sum_{i=1}^{2}f_{i}$$
(8)$$M\;\ddot{y}=\sum_{i=1}^{2}N_{i}-Mg+F_{m}sin(\omega t)$$

Furthermore, the reaction force $N_{i}$ can be calculated by writing the torques about the connecting point of the massless legs to the body.
(9)$$k\emptyset_{i}=-N_{i}L\cos\left(\beta_{i}\right)-f_{i}Lsin\left(\beta_{i}\right)i=1,2$$

In this case, the direction of the Coulomb friction force $f_{i}$ can be written in terms of the speed of the legs’ endpoints ${\dot{u}}_{i}$, which is the derivative of the endpoint position of the legs, $u_{i}$:(10)$$f_{i}=sgn\left({\dot{u}}_{i}\right)\mu N_{i}\;\;\;i=1,2$$
(11)$$u_{i}=x\pm \frac{l}{2}-L\cos\left(\beta_{i}\right)\;\;i=1,2$$
where l is the distance between the points connecting the two legs to the body, and L is the equivalent length of the leg measured from the contact point of the surface to the connecting point of body and leg.
(12)$${\dot{u}}_{i}=\dot{x}+L{\;\dot{\emptyset }}_{Di}\sin\left(\beta_{i}\right)\;\;\;i=1,2$$

The angle between the body and the legs is defined as $\beta_{i}$, which can be written in terms of the resting angle $\beta_{0}$, the angular displacement when the system is static with gravity $\emptyset_{Si}$, and the angular displacement of the dynamic system $\emptyset_{Di}$.
(13)$$\beta_{i}=\beta_{0}+\emptyset_{i}=\beta \;\;\;\;\;\;\;\;\;\;\;i=1,2$$
(14)$$\emptyset_{i}=\emptyset_{Di}+\emptyset_{Si}=\emptyset_{D}+\emptyset_{S}=\emptyset \;\;\;\;\;\;\;\;\;\;\;\;i=1,2$$

Further explanation and equations defining the parameters are presented in Appendix B.

#### 2.1.4. Case 4: Bristle-Bot with Jumping Feet

In this case, the maximum height of the legs was considered to determine when the legs left the ground (Figure 6).
(15)$$\delta =L+\frac{W_{M}}{2}$$

Although the legs returned to the resting angle when they were in the air, it was assumed that the bristle-bot jumped if Equation (16) was satisfied, as there are situations where $\beta_{i}$ is larger than the resting angle.
(16)y>δ

The leg with index i jumps if Equation (17) is satisfied.
(17)$$L\;sin\left(\beta_{i}\right)=L\;\;\;\;\;\;\;\;\;\;\;\;\;\;i=1,2$$

When the legs leave the ground, the reaction force is considered to be equal to zero.

#### 2.1.5. Case 5: Microrobot with Pitching Angle

In this case (Figure 7), a pitching angle was added to the DOFs of the system to mimic the realistic movement of the microrobot in the space defined by (x,y,α).
(18)$$M\;\ddot{x}=-\sum_{i=1}^{6}f_{i}-F_{m}sin(\alpha )$$
(19)$$M\;\ddot{y}=\sum_{i=1}^{6}N_{i}-Mg+F_{m}\sin\left(\omega t\right)cos(\alpha )$$

Considering the rigid body of the microrobot, the pitching angular acceleration can be written as follows:(20)$$\begin{array}{c} Moment\;of\;Inertia\times Pitching\;Angular\;Acceleration=\\ \sum Torque\;Due\;to\;Legs+\sum Torque\;Due\;to\;Eccentric\;Mass \end{array}$$
(21)$$\begin{array}{c} I_{zz}\;\ddot{\alpha }=F_{m}\sin\left(\omega t\right)\left(\frac{5}{2}l+d+d_{m}\right)-\sum_{i=1}^{6}f_{i}\left(\left(\frac{7}{2}-i\right)l\;sin\left(\alpha \right)+\frac{W_{M}}{2}cos\left(\alpha \right)\right)-\\ \sum_{i=1}^{6}N_{i}\left(\left(\frac{7}{2}-i\right)lcos\left(\alpha \right)-\frac{W_{M}}{2}sin\left(\alpha \right)\right)+k(\sum_{i=1}^{6}\emptyset_{i}) \end{array}$$
where $d_{m}$ is the distance from the center of mass of the eccentric mass to the front edge of the microrobot, d is the distance from the edges of the microrobot to the first connecting points of the legs, and $I_{zz}$ is the moment of inertia about the z axis.
(22)$$k\emptyset_{i}=-N_{i}L\cos\left(\beta_{i}\right)-f_{i}sin(\beta_{i})\;\;\;\;\;\;\;\;\;i=1,2,3,4,5,6$$
(23)$$f_{i}=sgn\left({\dot{u}}_{i}\right)\mu N_{i}\;\;\;\;\;\;\;\;\;\;\;\;\;\;\;\;i=1,2,3,4,5,6$$

Additional equations to derive the parameters of the above equations are reported in Appendix B.

#### 2.1.6. Case 6: Microrobot with Centrifugal Yaw Steering

In this case, the yaw DOF γ was added to the system to take into account the steering of the microrobot (Figure 8). It was assumed that the center of rotation was at the center of mass.

Considering the torques in this planar system, the angular acceleration in the yaw direction can be derived as follows:(24)$$\begin{array}{c} Moment\;of\;Inertia\times Yawing\;Angular\;Acceleration=\\ \sum Torque\;Due\;to\;Legs+\sum Torque\;Due\;to\;Eccentric\;Mass \end{array}$$
(25)$$I_{yy}\;\;\ddot{\gamma }=F_{m}\cos\left(\omega t\right)\left(\frac{5}{2}l+d+d_{m}\right)-\sum_{i=1}^{6}f_{li}r_{li}-\sum_{i=1}^{6}f_{ri}r_{ri}$$
where $I_{yy}$ is the moment of inertia about the y axis, $f_{li}$, $r_{li}$ and $f_{ri}$, $r_{ri}$ in the top view are the Coulomb friction force and the distance from the leg’s ground contact point to the center of mass of the left side and right side of the microrobot, respectively.
(26)$$f_{li}=sgn\dot{\left(\gamma \right)}\mu N_{li}\;\;i=1,2,3,4,5,6$$
(27)$$f_{ri}=sgn\dot{\left(\gamma \right)}\mu N_{ri}\;\;i=1,2,3,4,5,6$$

It was assumed that the only effect of coupling the previous case to this case was the calculated reaction force $N_{i}$ that was twice the value of the left $N_{li}$ and right $N_{ri}$ reaction forces:(28)$$N_{li}=N_{ri}=\frac{N_{i}}{2}\;\;i=1,2,3,4,5,6$$
(29)$$r_{li}=\sqrt[{2}]{{\left[\left(\frac{7}{2}-i\right)lcos\left(\alpha \right)-\frac{W_{M}}{2}\sin\left(\alpha \right)+L*\cos\left(\beta_{1}\right)\right]}^{2}+{\left[D_{l}\right]}^{2}}$$
where $D_{l}$ is the distance from the leg’s ground contact point to the midline of the body from the top view. As the microrobot was symmetric, based on the geometry of the system, the following equation applied to all legs:(30)$$r_{ri}=r_{li}\;\;i=1,2,3,4,5,6$$

### 2.2. Development

The assembly and development of small-scale robotics require precision, especially when dealing with robots whose movement mechanism depends on the vibration and properties of the materials used. In this section, the fabrication and assembly of four prototypes of microrobots for structural health monitoring are discussed.

#### 2.2.1. ESP32-CAM Mounted on HEXBUG-Nano

The first prototype was developed using HEXBUG-nano (Spin Master, Toronto, ON, Canada) for the movement mechanism and mounting an ESP32-CAM (Ai-Thinker, Shenzhen, China) with a 3.7 V 290 mAh lithium-ion battery glued to it using a double-sided mounting tape.

The most noticeable issue in using this system was the fact that the vibration of the unbalanced mass in the system made the images blurry.

Two approaches were used to solve this issue: one was a mechanical approach, which consisted in applying a soft material between the camera and the body to dampen the vibration, and the other involved using image stabilization techniques to reduce the noise in the images.

For this purpose, MATLAB R2023b software was used to capture multiple frames by accessing the IP address of the camera. Using these frames, an affine transformation between the features of the frame was determined, and, utilizing these transformations, a warped robust image was created to align the background planes that were not changed drastically.

#### 2.2.2. Microrobot Fabrication with Fused Deposition Modeling (FDM)

As additive manufacturing processes are capable of fabricating precise microrobots, they were used as the main method of development. In FDM printing, models are built layer by layer. The diameter of the microrobot legs was small, i.e., just 2 mm, and each layer had a relatively small surface area for adhesion, which led to a weak bonding between the layers and the notches around the legs, possibly resulting in breakage. Additionally, materials like PLA shrink upon cooling, affecting smaller-diameter cylinders even more significantly. This shrinkage generates internal stresses, making smaller cylinders more prone to breakage.

#### 2.2.3. Microrobot Fabrication with Silicone Molding

To create the microrobot legs using this technique, Boolean operations were used to derive the leg mold and vertically segment it into the sections A (lower segment in Figure 9) and B (upper segment in Figure 9) with SolidWorks 2022. Also, to guarantee the geometric alignment accuracy between A and B, three positioning pins were designed, corresponding to the protrusions in section A and the indentations in section B. Then, the molds were printed with an LCD 3D printer. Next, part A and part B of Smooth-On Dragon Skin 20 Mold-Making Silicone Rubber (Smooth-On, Macungie, PA, USA) were mixed and poured into the mold displayed in Figure 9. After overnight solidification, the legs of the microrobot were detached from the mold.

#### 2.2.4. MARSBot

Compared with FDM, LCD 3D printing excels in producing small components with higher resolution and lower layer height. Moreover, LCD printing accommodates a range of materials, from regular sturdy resins to flexible resins. Figure 10 shows several legs of the MARSBot developed using F80 Elastic Resin (Resione, Dongguan, China) with an LCD UV-cured 3D printer.

### 2.3. Control Interface

For structural health monitoring purposes, the microrobot needs to steer in a confined environment and send visual feedback to the operator for analysis. To achieve this, two interfaces were developed to control the movement of the microrobot. The first interface used a PC, and the second one used an AR headset (HoloLens 2) as an interface controller in a network connected to the HTC 5G hub.

The control scheme used for HRC is illustrated in Figure 11. With this method, humans constantly send commands by bending their fingers, which either streams the visual feedback or steers the microrobot. For steering the microrobot, different values were employed using the trial-and-error method to find the most intuitive value for the duration of the movement and ensure a smooth microrobot coordination by the human operator. In this control scheme, the human operator receives the visual feedback either through their eyes by inspecting the structure and the microrobot location or through the first-person view of the camera on the microrobot in AR, which assists in the coordination of the MARSBot movement to the designated area for inspection.

#### 2.3.1. PC Interface

As an initial attempt to develop an interface for the MARSBot, a Python script was run in JupyterLab 3.6.3 in Anaconda Navigator 2.4.0 to connect it to the ESP32-CAM through a Wi-Fi hotspot. For network communication, TCP was used. With this system, after uploading the server code on the ESP32-CAM via Arduino IDE 1.8.19 and powering on the MARSBot, the microrobot waits for the command from the client. With the JupyterLab script as the interface, three distinct characters on the keyboard can be typed that send commands to the MARSBot to either steer left or right or capture images and transmit them to the interface.

#### 2.3.2. Augmented Reality Interface

The connection between HoloLens 2 and ESP32 relies on UDP (User Datagram Protocol). There are two main elements on the user interface: the command sender and an image receiver. The command sender is used to send movement commands to ESP32’s vibration motor and live stream commands to the OV2640. By using the hand recognition API provided by the Mixed Reality Tool Kit (MRTK 2), the position of both palm and fingertips can be retrieved. HoloLens sends commands to ESP32 by calculating whether the angle of finger bending exceeds 100 degrees. The steps of the process are demonstrated in Figure 12. Meanwhile, due to potential signal instability, to prevent thread blocking, asynchronous transmission is used during image transfer.

## 3. Results

In this section, the prototypes obtained using each development method and their features are described. Then, based on the properties of the most efficient prototype, simulations of the dynamic equation are presented. Finally, the interfaces and their function are displayed and elaborated.

### 3.1. Prototypes

#### 3.1.1. ESP32-CAM Mounted on HEXBUG-Nano

The first prototype was an ESP32-CAM mounted with a battery on top of the HEXBUG-nano, as displayed in Figure 13. This microrobot moves based on the movement of the HEXBUG-nano and can be used for pipeline inspection or general SHM inspections. However, its direction cannot be controlled in the current configuration.

Figure 14 shows the results of image stabilization using point feature matching. As shown in the figure, the object in the average-corrected-sequence image is clearer than that in the average-raw-input image.

In this image, MATLAB video stabilization based on point feature matching was utilized [34]. In this application, the IP address of the ESP32-CAM was accessed, and 25 frames were captured by the camera. The images were converted to grayscale to improve the speed of the algorithm. Next, the algorithm used features from the accelerated segment test (FAST [35]) corner detection algorithm to detect the keypoints and the fast retina keypoints (FREAK [36]) to extract the descriptor around them. The corresponding points among the descriptors obtained were found using the matchFeature function in MATLAB. Using the M-estimator sample consensus (MSAC [37,38]) algorithm, an affine transformation between the corresponding points was estimated. Then, a similarity transformation matrix was returned by using the extracted scale, rotation, and translation from the affine transformation matrix. Using the previous similarity function, a cumulative transformation was defined and updated at each iteration, by capturing a new frame. Finally, the imwarp function in MATLAB was utilized to incorporate the cumulative transformation and warp the image to align it with the first image captured by the camera.

#### 3.1.2. Microrobot with Extrusion-Based 3D-Printed Legs

This microrobot has two key features: the ESP32-CAM and the legs. Figure 15 displays the microrobot legs that were 3D-printed using FDM with PLA materials. This microrobot has rigid and brittle legs. Using this method, the obtained legs can easily brake and have high stiffness, which does not result in the best movement performance of the microrobot.

#### 3.1.3. Microrobot with Silicone-Molded Legs

A microrobot with silicone-molded legs is depicted in Figure 16. Due to the low stiffness of the material, legs with the same diameter as that in the previous design could not bear the weight of the ESP32-CAM. To solve this issue, the diameter of the legs was increased. However, this led to bulky legs that slowed down the movement of the microrobot.

#### 3.1.4. MARSBot

The MARSBot consists of the ESP32-CAM, a battery, a DC motor with eccentric mass extracted from HEXBUG-nano Nitro, a board mount, and legs. Figure 17 shows the MARSBot with LCD-printed mount and legs. The ESP32-CAM mount serves to connect the ESP32-CAM board to the legs; basic white resin was utilized for its fabrication. A rubber-like resin was used for the legs to achieve a certain level of flexibility and stiffness.

For a comparison between the prototypes, an experiment was conducted using a pipe with a length of 691.5 mm. In this experiment, each microrobot was placed at the beginning of the pipe, and the time that it required to reach the end of the pipe was determined to calculate the speed of each microrobot.

Table 1 shows the speed, dimensions, weight, and Durometer of the legs. It is worth noting that in prototype number 4, the eccentric mass had lower weight than in the other cases.

Several more microrobots were developed to obtain these final four prototypes. Considering the hardness of the legs of the microrobot, among the four materials used for the prototypes, the material employed for silicone molding had the lowest durometer. Therefore, in order to carry the weight of the EPS32-CAM, the design was changed, and the diameter of the legs was increased. This not only led to a higher mass but also increased the contact of the legs with the surface, which raised the friction force. On the other hand, the FDM 3D-printed legs had the highest durometer; so, their diameter could be reduced. However, they were brittle and were not sufficiently flexible to facilitate the movement of the microrobot.

It was noticed that among all features, two key features were crucial for building an effective microrobot based on the bristle-bot movement mechanism. The first feature is the weight of the microrobot compared to the eccentric mass, and the second feature regards the characteristics of the legs of the microrobot. Depending on the method and the materials, the prototype requires legs with different diameters to bear the weight of the microrobot, which results in either a heavier or a lighter system. Therefore, when selecting the eccentric mass, the choice of materials and method depends on first ensuring that the microrobot will be capable of carrying its weight, while optimizing parameters such as the frequency of the eccentric mass, the mass of the system, and the stiffness of the legs, so that the system approaches the resonance frequency, for maximum performance of the microrobot.

### 3.2. Simulation

In this section, the simulation of the motion of the system is described. MATLAB version 2023b ODE45 function [39] running on a PC with Intel(R) Core (TM) i7-10870H CPU@2.20 GHz was used to numerically solve the equation of motion of the system.

#### 3.2.1. Parameters

In this simulation, the properties of the MARSBot were considered as the basis for all simulations, except when otherwise indicated. The parameters used in these simulations were M=22.35 g, $e=1.18\times 10^{-3}\;m$, $g=9.81\;\frac{m}{s^{2}}$, $l=6.76\times 10^{-3}\;m$, $d=3\times 10^{-3}\;m$, $d_{m}=2.588\times 10^{-3}\;m$, $W_{m}=13.720\times 10^{-3}\;m$, $L=0.0142\;m,\;{\;b}_{0}=0.0142\;rad$, $\emptyset_{s}=-0.1053\;rad$, $I_{zz}=3.2492\times 10^{-6}\;kg\cdot m^{2}$, and $I_{yy}=5.1500\times 10^{-6}\;kg\cdot m^{2}$.

The parameters that were not known, such as ω, k, and μ , were determined as follows.

For the determination of ω, the accelerometer ADXL335 was attached to the MARSBot and connected to Arduino Uno. The motor was turned on five times, each time for 2 s. Figure 18 depicts the analog data for the X, Y, and Z axes.

Using the analog output data, the calculated frequency based on the peaks in the graphs resulted to be 6.99 Hz and was used for the determination of the eccentric mass angular velocity.

To find the mass of the eccentric mass, the material was assumed to be alloy steel. Considering the density of alloy steel and the approximate dimensions of the eccentric mass, we determined that m=0.35 g.

An additional important parameter is the stiffness k. In order to determine this parameter, three masses were added to the system, and the displacement was measured using an optical comparator. Table 2 shows the force and displacement values after adding each mass on the top of the MARSBot.

Figure 19 shows a quadratic curve fitted to the data from Table 2; the tangent line to the starting point of the curve, where the main operational range of the MARSBot is, was drawn, and the stiffness was determined by calculating the slope of the tangent line, which was 1315.51 N/m. In addition, the linear stiffness needed to be converted to the rotational stiffness using the moment arm, according to $k=(\frac{1}{n})1315.51{\left(0.0025\right)}^{2}$, in which n is the number of legs.

Moreover, the stiffness could also be determined by using the formula $k=\frac{EA}{L}$ and substituting the values of the Young’s modulus E from the manufacturer’s data (Table 3).

The slope of the tangent line was the closest value to the theoretical stiffness of 1579 N/m. This was due to the fact that the geometry changed as we applied more load to the system.

#### 3.2.2. Simulation of Case 1

For running the first case simulation, the system was assumed to be on an anisotropic surface with the properties of Scots pine wood [40], i.e., μ+=0.17, and μ−=0.08, and the initial conditions were set to (0, 0).

Figure 20 depicts the x-direction displacement of the system as a function of time. As it is shown in the graph, the system moved in one direction, while the eccentric mass rotation changed the speed of the system.

#### 3.2.3. Simulation of Case 2

In the second case, the initial conditions were set to (0.001, 0.001). As displayed in Figure 21, the system performed a typical oscillation around the equilibrium point.

#### 3.2.4. Simulation of Cases 3 and 4

In this simulation, the friction coefficient of the PVC surface was calculated using the tangent of the angle of repose of the inclined plane when the MARSBot started to slide. Using this method, the friction coefficient was determined to be μ=0.257.

The initial conditions were set to (0, 0, 0.0209, 0). Figure 22 depicts the horizontal location of the center of mass of the bristle-bot as a function of time. In this graph, it is shown that the system moved in a positive direction, oscillating around a positive-slope line.

The system fluctuated in the vertical direction as expected, around the equilibrium point, as depicted in Figure 23.

Moreover, Figure 24 shows the position of the feet of the bristle-bot in the horizontal direction with respect to time. The feet followed a cyclic pattern around a positive-slope line.

#### 3.2.5. Simulation of Case 5

In this case, the pitching angle was added to the DOFs of the system. As the equation was more computationally expensive, the absolute tolerance was increased and set to $10^{-4}$ to expedite the process. The initial conditions were set to (0, 0, 0.029, 0, 0.001, 0).

Figure 25 and Figure 26 show approximately the same oscillatory pattern as in the previous case.

The pitching angular displacement decreased and approached the equilibrium point, as displayed in Figure 27. The pattern of the angular displacement was periodic. Nevertheless, a jumping behavior is evident in the zoomed view of Figure 27.

Figure 28 shows a rising behavior of the position of the feet, while the jumping pattern from Figure 27 is reproduced in the zoomed view of this plot.

#### 3.2.6. Simulation of Case 6

In addition to the pitching angular displacement, in Case 6, the yawing angular displacement was added to the system. For solving this equation, due to its complexity, the time span was reduced to 0.714 s with relative and absolute tolerance of $10^{-3}$. This configuration enabled the solver to obtain results in a reasonable time.

Figure 29 plots the yaw angular displacement of the MARSBot. The graph displays small vibrations and jumps whenever the eccentric mass was at the right position to steer the microrobot. The value of the variable shows that the yaw angular displacement increased, indicating the steering behavior of the MARSBot.

### 3.3. Interface Control Panel

In this section, the user interfaces that were developed for the MARSBot are depicted and described for structural health monitoring applications.

#### 3.3.1. PC Interface

The interface in JupyterLab is shown in Figure 30. In this interface, the commands marked as ‘g’, ‘j’, and ‘h’ allowed steering to the left, steering to the right, and receiving visual feedback, respectively. These functions were built into the MARSBot code and were transmitted to the microrobot by typing them on the keyboard of the PC.

When deriving the equations of motion, multiple assumptions were made to simplify the motion of the microrobot for dynamic modeling. To find the deviation of the experimental results for the prototype from the simulation results for the dynamic model, multiple experiments were conducted on a gridded piece of paper composed of squares with 5 mm sides. In these experiments, the PC interface was used, and the MARSBot was programmed to run for 1 s after receiving a command. As the forward distance traveled by the microrobot varied, the experiment was repeated nine times, and the distance traveled was 20 mm in five times, with about 2.5 mm accuracy. For comparison, the simulation of Case 5 was repeated, measuring the coefficient of friction of the paper with the same procedure described in Section 3.2.4. The forward distance traveled in this simulation was determined to be 20.6 mm and was compared to the experimental result, as shown in Figure 31.

#### 3.3.2. AR Interface

The control panel used has four buttons arranged in the following order, from top to bottom: Controller, Stream, Settings, and Back. When hitting the Controller or Stream button, HoloLens searches the local network for UDP connections to the ESP32-CAM. If there are no suitable UDP connections in the local network, the Controller and Stream buttons turn grey, and the user can rescan this local network within the Settings button. After successfully finding UDP connections to the ESP32-CAM, by hitting the Controller button, three green circular indicators float above the fingertips, as shown in Figure 32. Upon detecting finger bending exceeding 100 degrees, the corresponding commands are sent to the ESP32-CAM, and the indicator turns red, as shown in Figure 32. The Stream button activates a screen floating around 10 m in front of the user to display images from the ESP32-CAM in real time. The screen size and distance from the user can also be adjusted in the settings. The Back button simply terminates the UDP connections and stops the program.

### 3.4. Case Studies Involving the MARSBot

#### 3.4.1. Slotted Strut Channel

The Unistrut channels were utilized as ceiling grids and mounting and supporting structures. As they have narrow passages and are used in hard-to-reach areas, it is not very easy to inspect them.

Figure 33 shows the MARSBot inspecting a Unistrut channel.

The MARSBot could move and fit on either side of the Unistrut channel. This experiment is depicted in Figure 34, showing the visual feedback provided for the connections and the slotted side of the channel.

#### 3.4.2. AR Steering

In this process, the operator wore the AR headset (HoloLens 2), and the MARSBot steered whenever the operator bent his fingers, as displayed in Figure 35. Using this method, operators can receive first-person visual AR feedback from the interior of the Unistrut channel, while viewing the outside of the structure with their eyes and steering the MARSBot from the third-person view to facilitate the inspection and human–robot collaboration.

## 4. Discussion

AR inspection of structures assists the operator in detecting damage by looking at the structure from the outside and using the AR overlay for inspecting the inside of the structure. This method allows for a better understanding of where defects are located for future maintenance and repair. In narrow passages, confined spaces, underground infrastructure, and ceilings, only small-scale robotics systems can be used; therefore, a microrobot that can steer according to the command of the operator wearing the AR headset will facilitate and speed up the process of inspection.

### 4.1. Discussion of the Results

In the development of the prototypes, both FDM and LCD printing could print flexible materials. However, due to the small cross section of the legs, there was insufficient adhesive strength between the FDM-printed layers. Therefore, given that the height of the LCD-printed layers was only 0.05 mm (in contrast to, typically, 0.2 mm for the FDM-printed layers), LCD appeared to be more suitable for printing these small legs. Additionally, during LCD printing, each layer could be cured within 5 s, which resulted in a much shorter LCD printing time compared to that required for FDM printing.

The comparison between the prototypes demonstrated the higher speed of prototype #4 (MARSBot), with a durometer of 72, compared to prototype #1 (HEXBUG-nano), with a durometer of 64, and to prototype #3, with bulkier and more flexible legs and a durometer of 44. A comparison of the speeds of the three prototypes showed a great difference between the speed of prototype #3 and those of the other two prototypes. Although this could be caused by the lower eccentric mass in prototype #3, the high flexibility of its legs exacerbated the need for a higher diameter to carry the microrobot weight, which may also be the reason for the low speed of this prototype.

The results of the simulation demonstrated the movement mechanism of the microrobot. In each period, despite the center of mass moving back and forth, the microrobot moved in the forward direction, and the pitching angle in the modeling demonstrated to have a positive effect on increasing the forward displacement. Moreover, including the yawing angle in the modeling evidently explains the steering behavior of the microrobot.

The interfaces, starting from the PC interface and evolving to a more user-friendly and practical interface version in AR, displayed an intuitive way of controlling the MARSBot, as demonstrated in the Appendix A. Using this method, the operator can use the MARSBot in confined spaces and inspect narrow structures without any previous training. Using the microrobot and the AR headset in wireless networks creates a lag between the user’s hand command, the AR headset detecting the command, and the microrobot movement. For the best results in microrobot operation, this lag should be minimized. An experiment was conducted on a wooden surface, with no obstacle surrounding it, to measure the lag when employing the indicated network by using a digital clock on the display and recording the procedure through the AR headset. The results showed a lag of 128 ms from the time when the finger was bent to the time the AR headset detected the command and turned the touched button red. Then, it took about 240 ms for the microrobot to receive the command from the AR headset and start its movement. Moreover, it should also be noted that the material of the structure intended for inspection might interfere with the transmission of wireless signals.

### 4.2. Comparative Analysis

Considering the recent advances in the development of microrobots, a comparison was made between some of the prevalent features of recent microrobots and those of the MARSBot.

The microrobots using a piezoelectric actuator usually require high voltage, namely, 40 V for a microrobot with a length of 12 mm and a weight of 0.2 g [16]. Since it is hard to provide high voltages with an on-board power source without significantly increasing the weight of the microrobot, these microrobots are usually wired to the power source. This is also true for ionocrafts, as even those that weigh 67 mg and are 2 cm long require 2000 V [19] to liftoff. On the other hand, the MARSBot with the weight of 22 g and the length of 4.7 cm only requires 3.3 V both as an input to the microcontroller and as an output to the DC motor to operate, which can be provided by a regular small-scale on-board power supply.

Moreover, there are magnetically driven microrobots with a length of 2.2 mm that are wireless, in the sense that they operate untethered. However, using the 3D Helmholtz coil system requires 10 A of current to generate 20 mT of uniform magnetic field for controlling it [22]. This requirement will create a constraint on the power supply and on the location of the microrobot. Nevertheless, the MARSBot only requires a wireless connection to receive the control commands and send visual feedback.

In addition to the common benefits of monitoring in hazardous conditions that robotic systems, including expensive mobile robots, quadruped robot dogs, namely, Spot, by Boston Dynamics, drones, and wall-climbing robots, provide compared to human inspection, other advantages are offered, e.g., for areas of SHM, specifically, confined space monitoring, which lack practical approaches with robotic systems. Monitoring hard-to-reach areas in SHM is important, as some defects usually start to appear in areas that are not visible during routine inspections. This task may require the disassembly of the structures.

Visual inspection is one of the major means of detecting damage in SHM. The current methods for performing this task use an endoscope and similar systems, whose performance is limited by the length of the tether wire and which are difficult to coordinate. There are specific bristle-bots designed for the monitoring of pipelines using tethered approaches. However, most of the novel microrobots were produced as a proof of concept for their movement mechanism and lack either the main means of inspection or practicality, to be used for SHM.

The proposed method to monitor hard-to-reach areas is based on the use of the MARSBot. Not only it is wireless with an onboard power source, but also it is small, providing accessibility to structures that could not be inspected before. It is worth noting that it was designed using commonly available low-cost microcontroller and 3D printing methods, with affordable equipment, which facilitates its replication. Furthermore, as the MARSBot provides visual feedback to the AR headset-wearing operator, it improves the traditional visual inspection made using the naked eye, as the operator can view the outside of a structure with the eyes and the inside of confined structures through the MARSBot first-person view.

### 4.3. Future Trends

Future research directions regarding the MARSBot include:-Dynamic modeling of the 3D model with the comprehensive coupling effect.-Modifying the legs of the microrobot for a variety of applications.-Using different motor types and the addition of extra motors while varying the frequencies for precision control.-Enhancing the precision, tuning, and control for unbiased forward movement.-Developing advanced control strategies allowing the MARSbot to move toward the designated location determined by the AR headset user.-Incorporating artificial intelligence and machine learning algorithms in edge processors to detect objects or defects in structures and use control schemes to move the MARSBot toward them for a thorough inspection.-Tuning the stiffness and mass of the system and the rotor frequency based on the natural frequency of the system for optimal performance.-Adjusting the frequency of the eccentric mass to allow the system to move in a backward direction.

## Figures and Tables

**Figure 1 micromachines-15-00202-f001:**
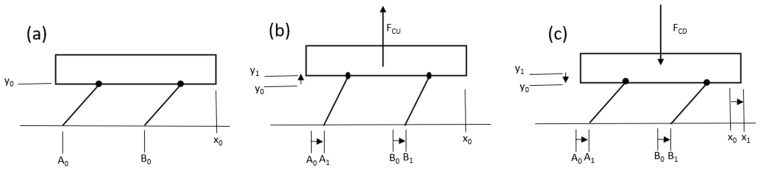
Bristle-bot movement mechanism when the eccentric mass is located in the center of mass, which shows the eccentric mass movement process that leads to the movement of legs and body (**a**) original position of the body, (**b**) elevated position, and (**c**) return to original height with forward movement.

**Figure 2 micromachines-15-00202-f002:**
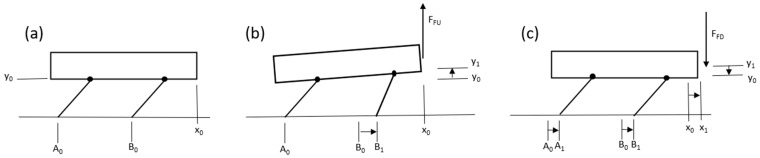
Bristle-bot movement mechanism when the eccentric mass is located at the front of the bot; in this case, the eccentric mass movement results in front leg movement, followed by body and rear leg movement (**a**) original position of the body, (**b**) pitching up of front of body, and (**c**) return of body to original pitch angle with forward movement.

**Figure 3 micromachines-15-00202-f003:**
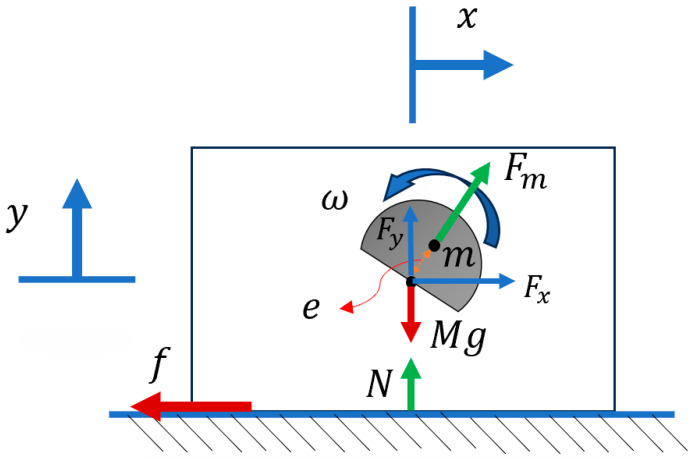
Eccentric mass on a system with two DOFs on a surface with Coulomb friction, showing the basic mechanism of movement of the bristle-bot.

**Figure 4 micromachines-15-00202-f004:**
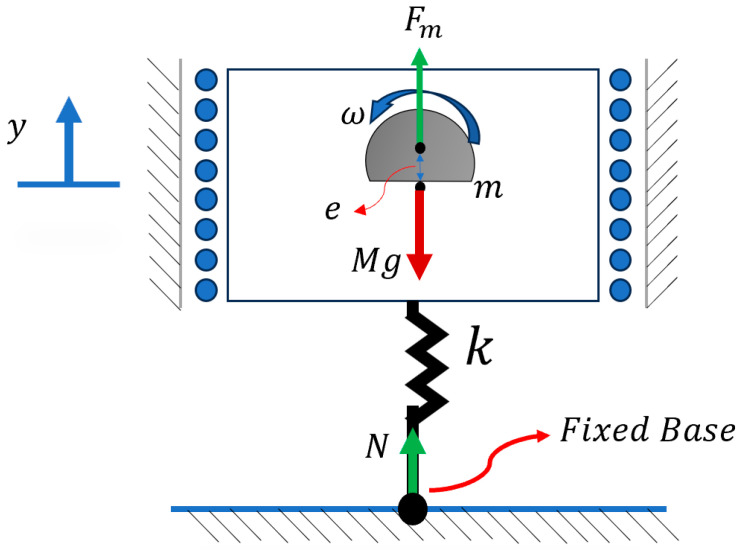
Eccentric mass with one DOF on a spring, displaying the simplified version of the bristle-bot.

**Figure 5 micromachines-15-00202-f005:**
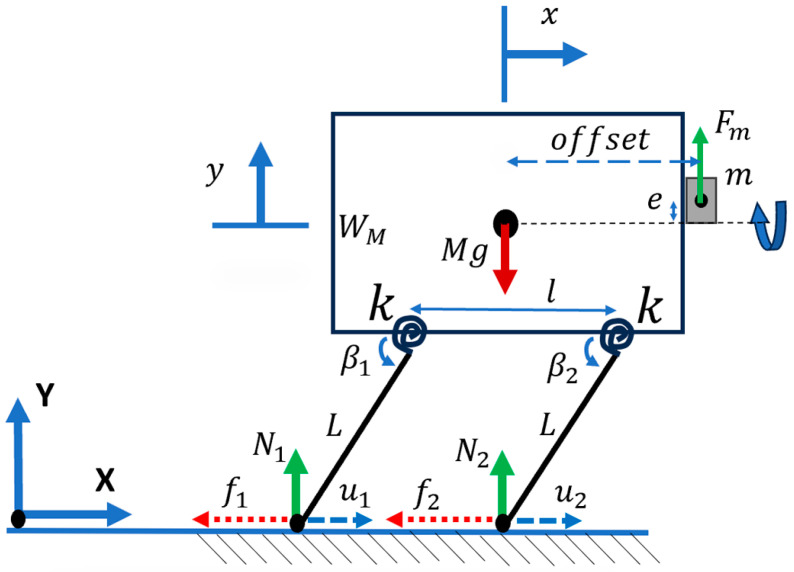
Bristle-bot with two offset legs including the main features of the bristle-bot movement mechanism.

**Figure 6 micromachines-15-00202-f006:**
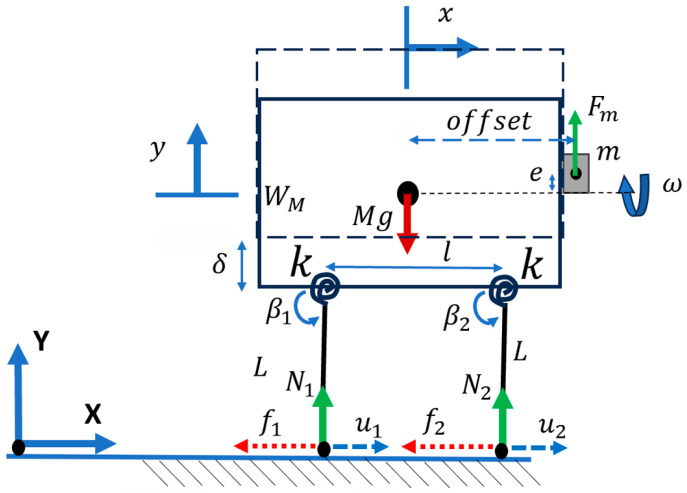
Microrobot with 2 legs at the maximum height when they are about to leave the ground.

**Figure 7 micromachines-15-00202-f007:**
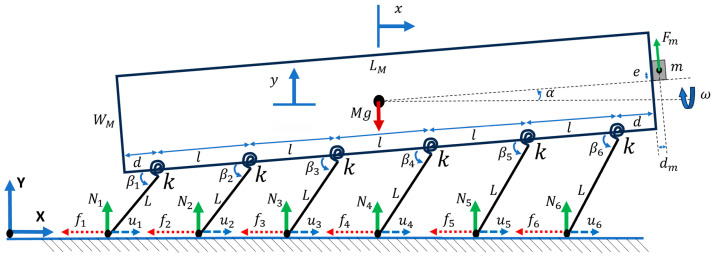
Addition of a pitching angle to the microrobot with 3 DOFs and offset legs.

**Figure 8 micromachines-15-00202-f008:**
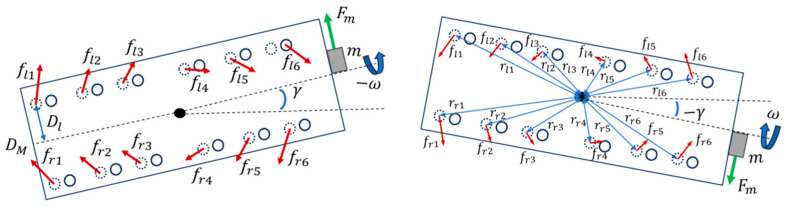
Top view of the microrobot with steering angle dependent on the direction of the angular velocity.

**Figure 9 micromachines-15-00202-f009:**
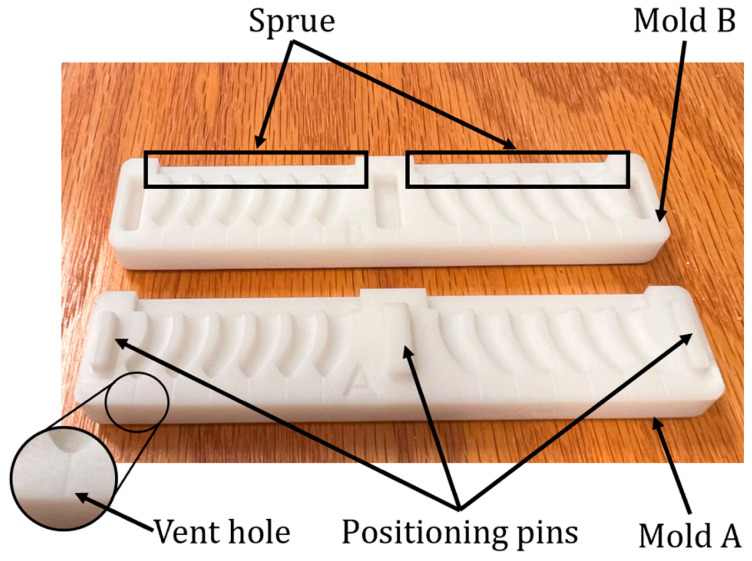
Two segments of the mold used for the fabrication of the microrobot legs with silicone rubber.

**Figure 10 micromachines-15-00202-f010:**
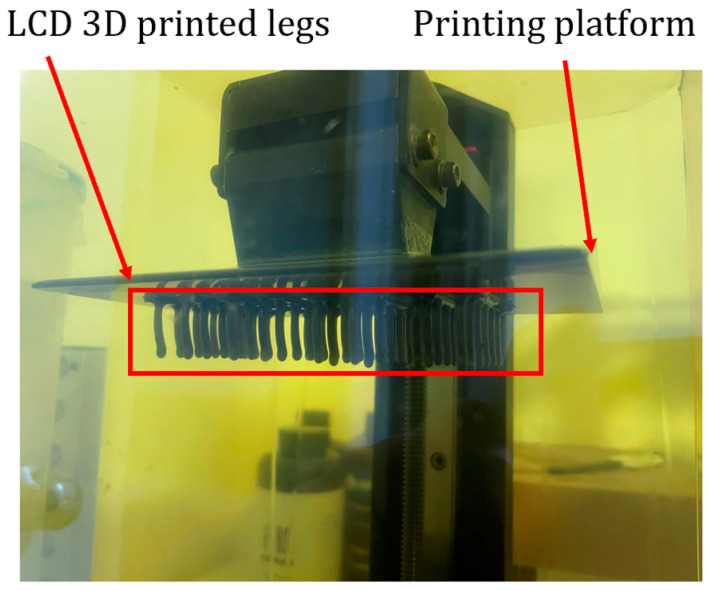
LCD UV-cured 3D printer with a series of MARSBot legs in the printing platform.

**Figure 11 micromachines-15-00202-f011:**
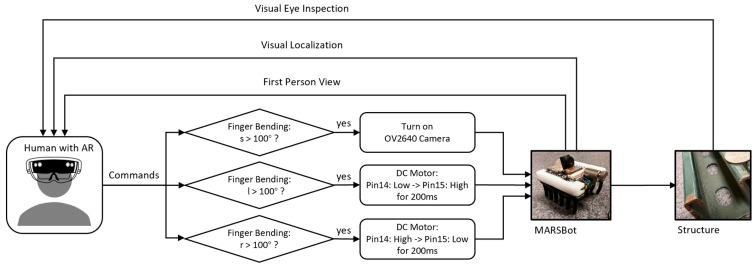
HRC control scheme for structural health monitoring using finger movements for an intuitive control of the microrobot.

**Figure 12 micromachines-15-00202-f012:**
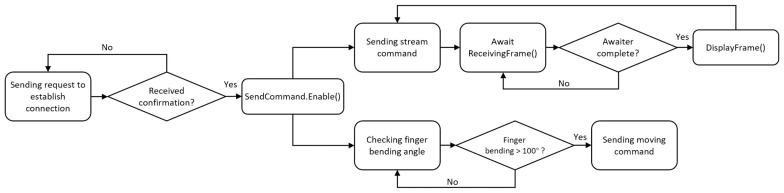
Flowchart of the steps for sending commands and receiving visual feedback through the augmented reality user interface.

**Figure 13 micromachines-15-00202-f013:**
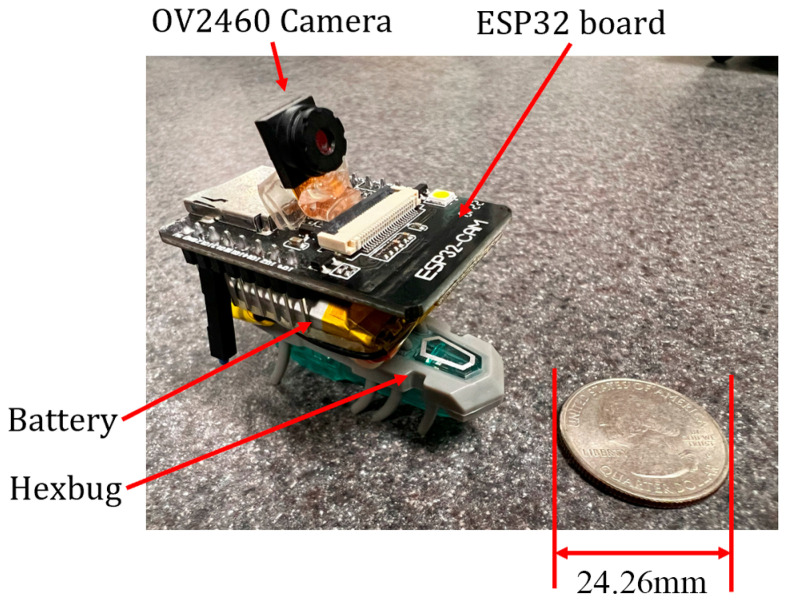
ESP32-CAM mounted on the HEXBUG-nano for structural health monitoring with random direction movement.

**Figure 14 micromachines-15-00202-f014:**
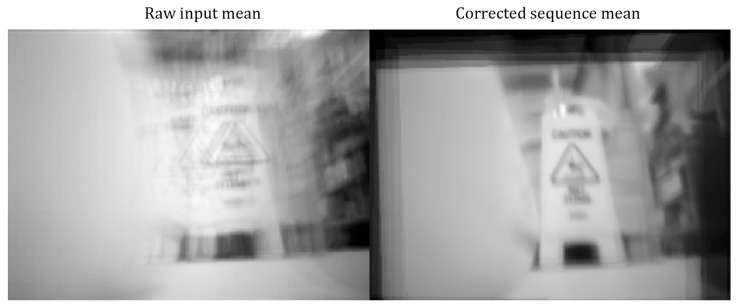
Image stabilization through point feature matching for reducing the effects of the microrobot vibrations on visual feedback.

**Figure 15 micromachines-15-00202-f015:**
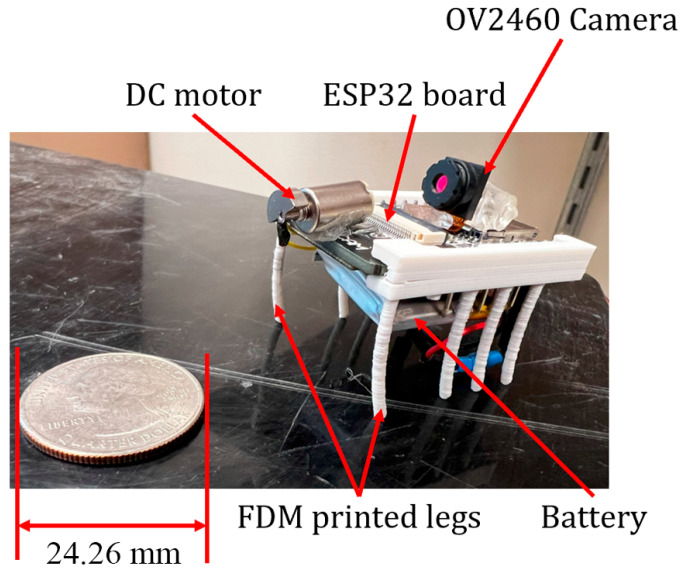
Microrobot mounted on legs that were 3D-printed using PLA materials.

**Figure 16 micromachines-15-00202-f016:**
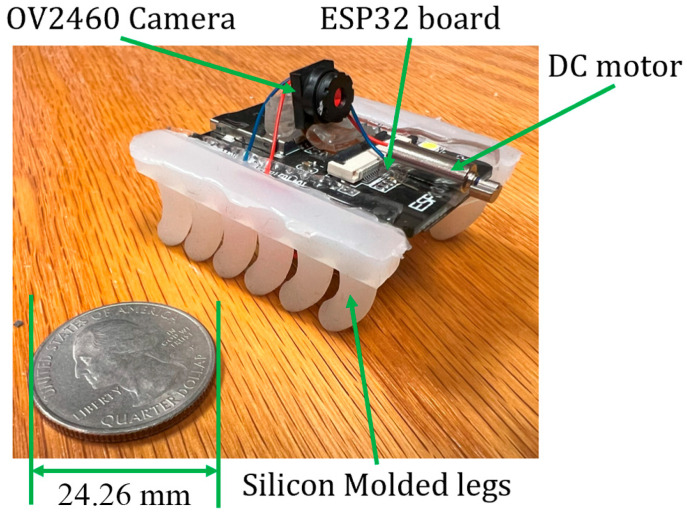
Microrobot mounted on silicon legs developed using the molding process.

**Figure 17 micromachines-15-00202-f017:**
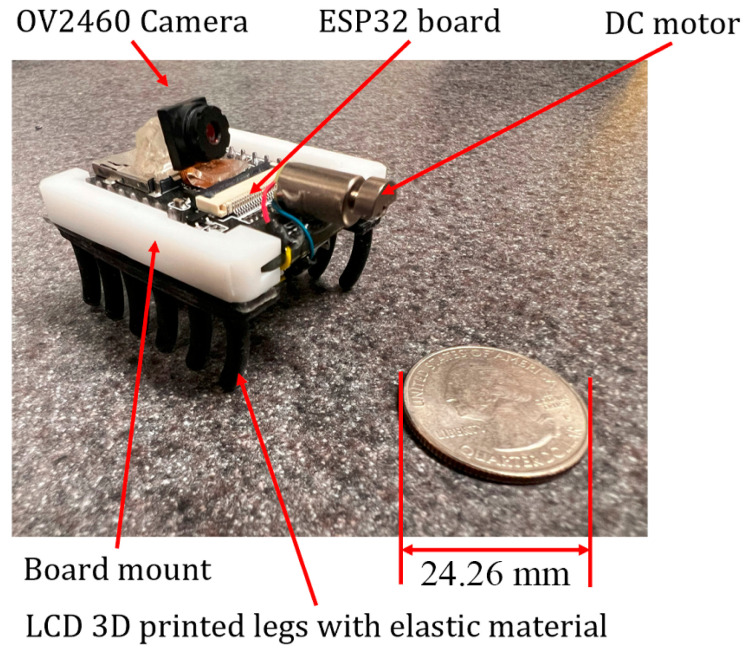
Microrobot with rubber-like UV-cured 3D-printed legs and board mount.

**Figure 18 micromachines-15-00202-f018:**
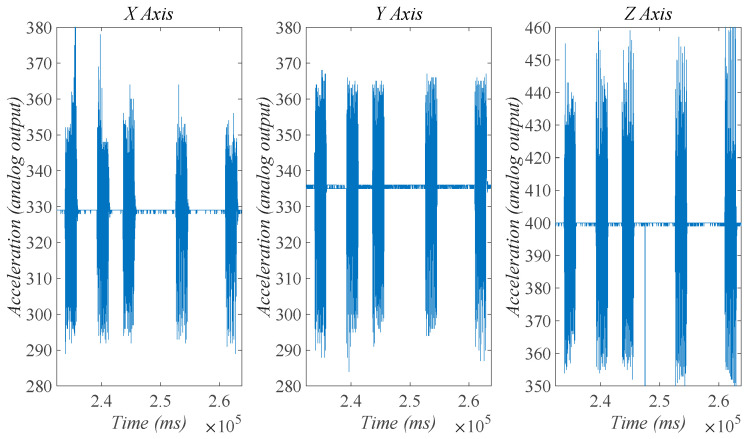
Analog output of the accelerometer during 5 successive 2 s rotations of the eccentric mass captured in the X, Y, and Z directions.

**Figure 19 micromachines-15-00202-f019:**
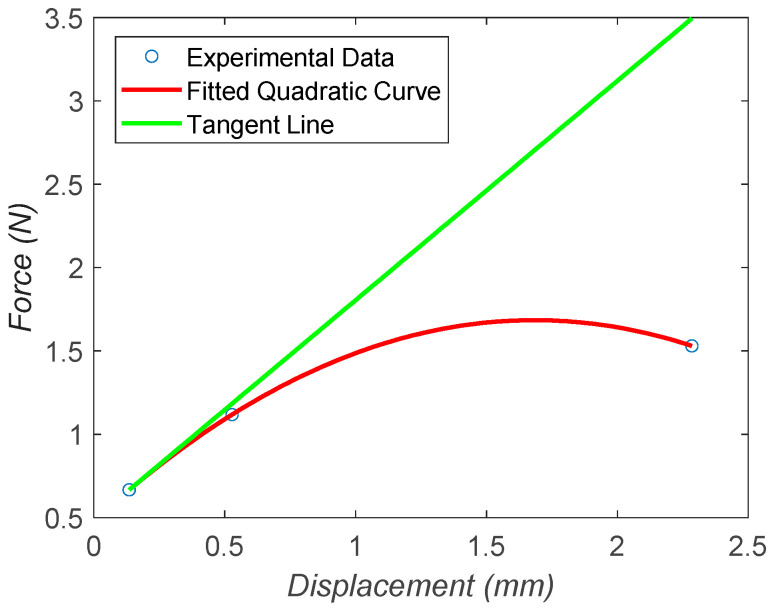
Force–displacement graph for the approximation of the stiffness at the starting point, based on Table 2 data.

**Figure 20 micromachines-15-00202-f020:**
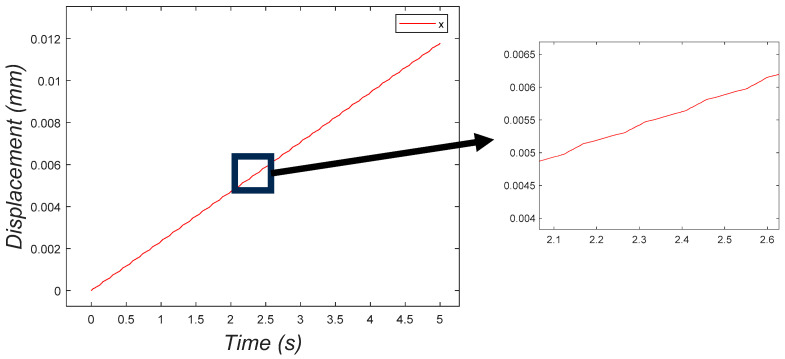
Displacement of the system with an eccentric mass in the x direction.

**Figure 21 micromachines-15-00202-f021:**
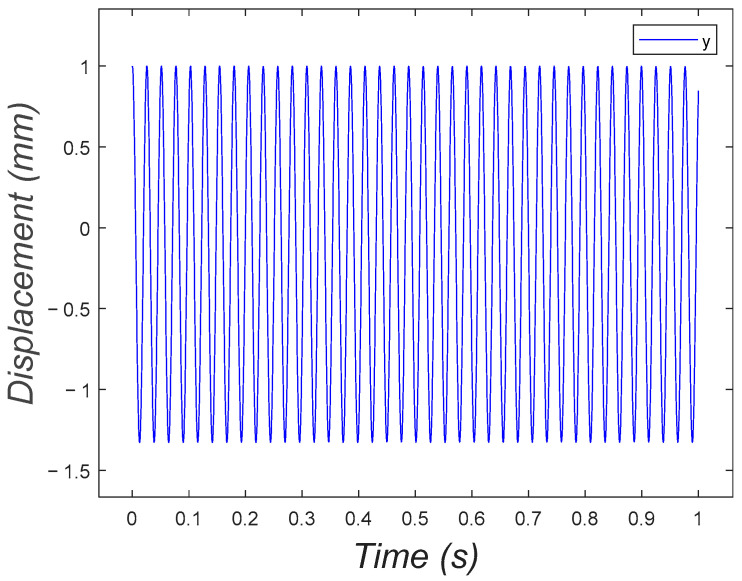
Oscillation of the single-DOF eccentric mass system on a spring around the equilibrium point.

**Figure 22 micromachines-15-00202-f022:**
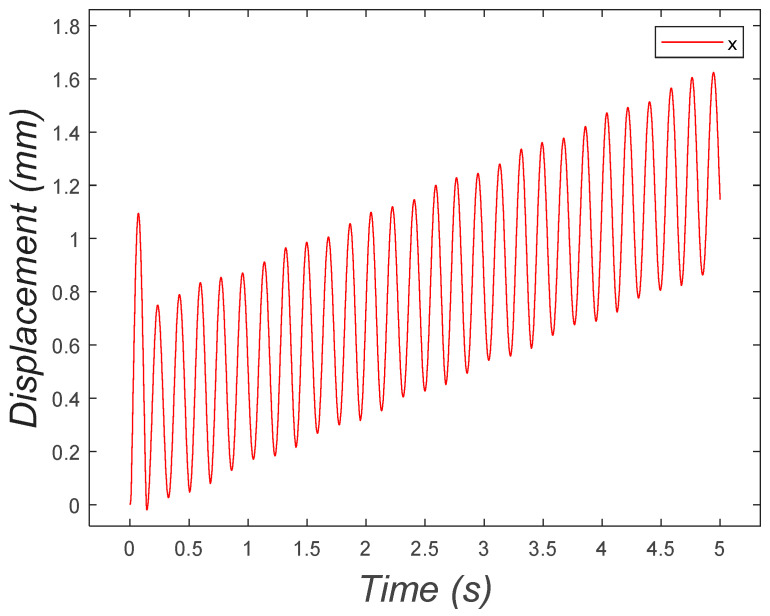
Horizontal position of the two-DOF microrobot with two legs oscillating around a positive-slope line.

**Figure 23 micromachines-15-00202-f023:**
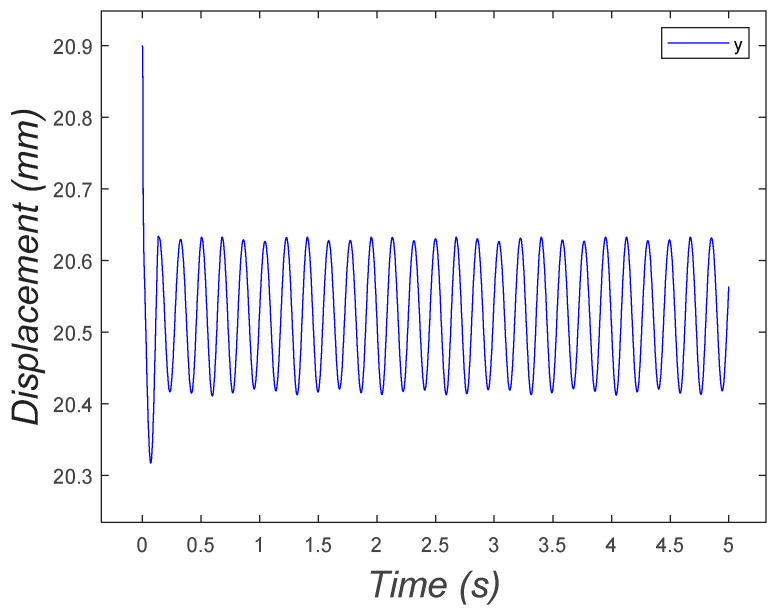
Vertical position of the two-DOF microrobot with two legs fluctuating around the equilibrium point.

**Figure 24 micromachines-15-00202-f024:**
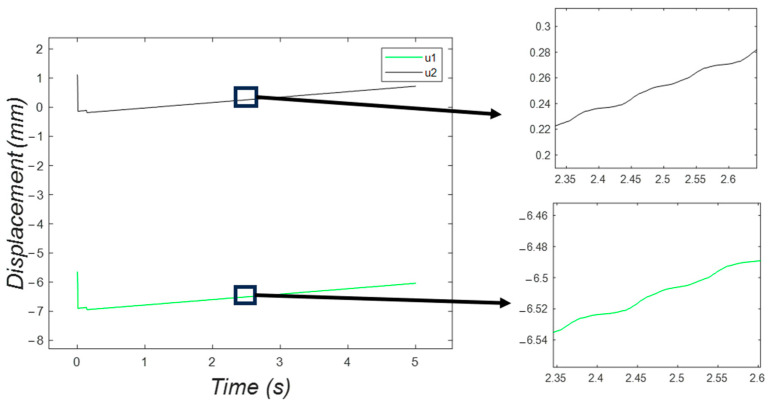
The horizontal position of the feet in the two-DOF system, fluctuating following a rising trend.

**Figure 25 micromachines-15-00202-f025:**
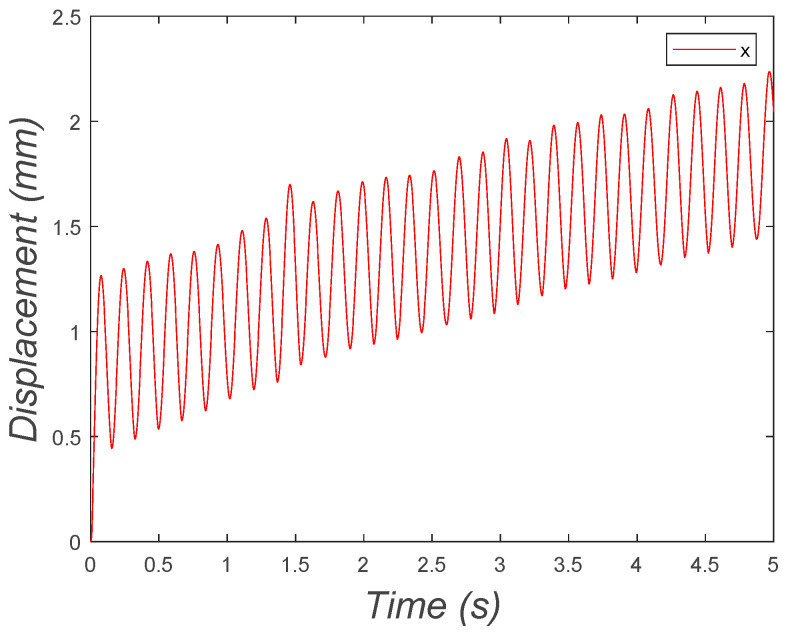
Horizontal position of the three-DOF microrobot with six legs oscillating around a positive-slope line.

**Figure 26 micromachines-15-00202-f026:**
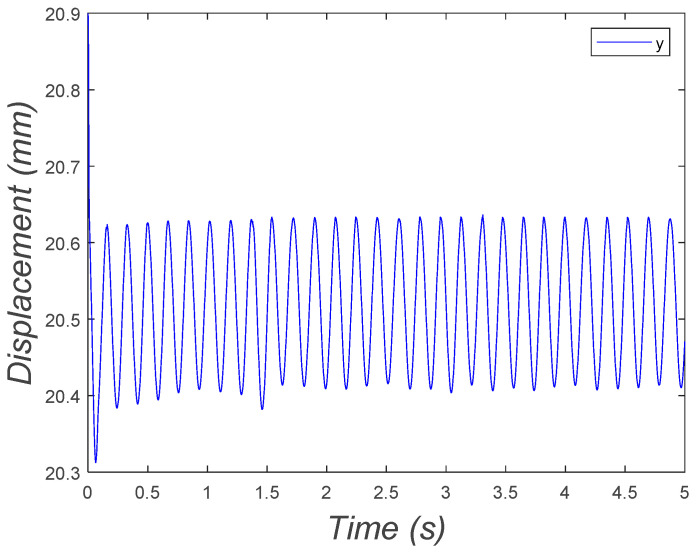
Vertical position of the three-DOF microrobot with six legs fluctuating around the equilibrium point.

**Figure 27 micromachines-15-00202-f027:**
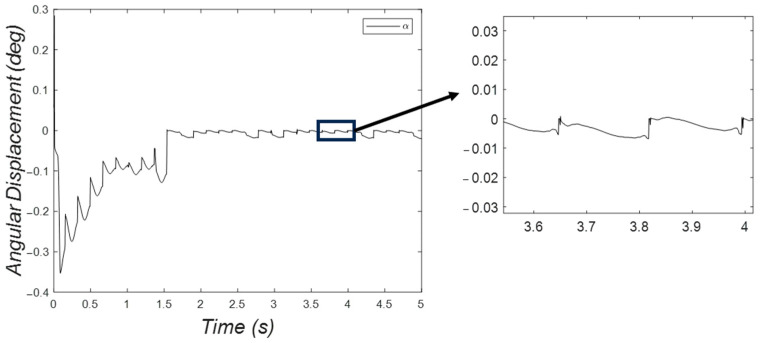
Pitching angular displacement of the three-DOF microrobot with six legs decreased from the initial condition to a small oscillatory behavior around the equilibrium point.

**Figure 28 micromachines-15-00202-f028:**
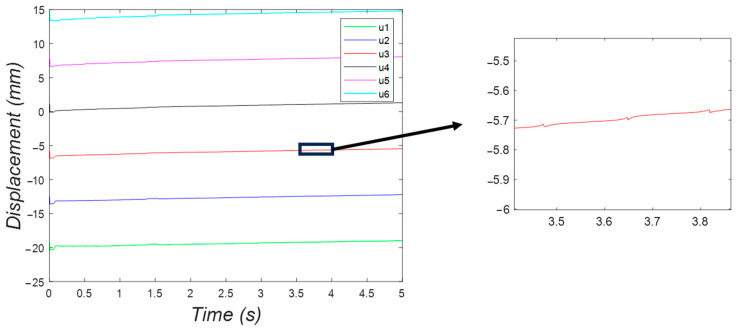
Horizontal displacement of the six feet in the three-DOF system, with a rising fluctuating behavior.

**Figure 29 micromachines-15-00202-f029:**
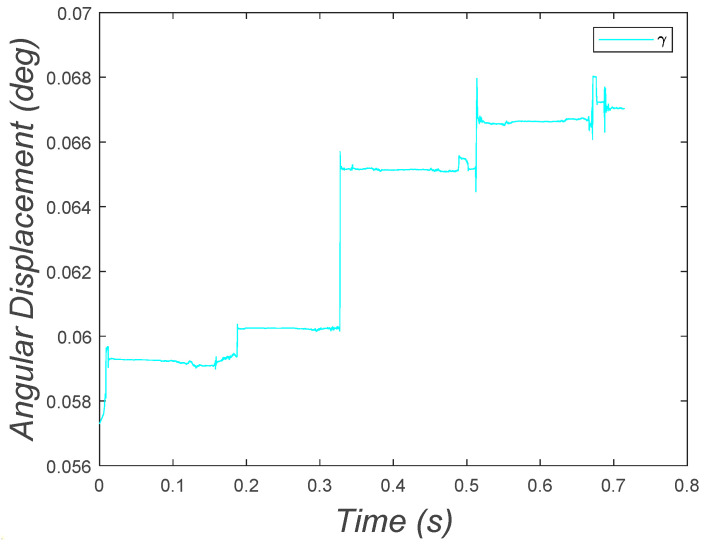
Yaw angular displacement of the four-DOF system, showing the increasing of the yaw steering of the MARSBot.

**Figure 30 micromachines-15-00202-f030:**
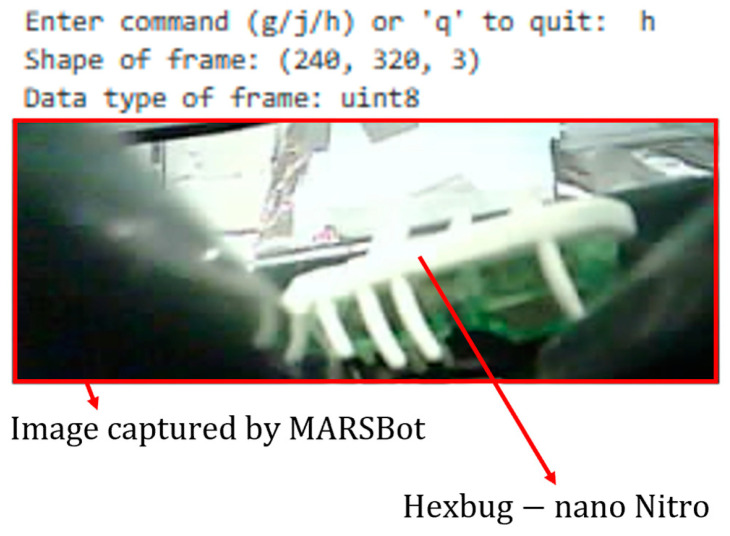
Captured image of the HEXBUG-nano Nitro by the MARSBot in Unistrut channels using the PC interface.

**Figure 31 micromachines-15-00202-f031:**
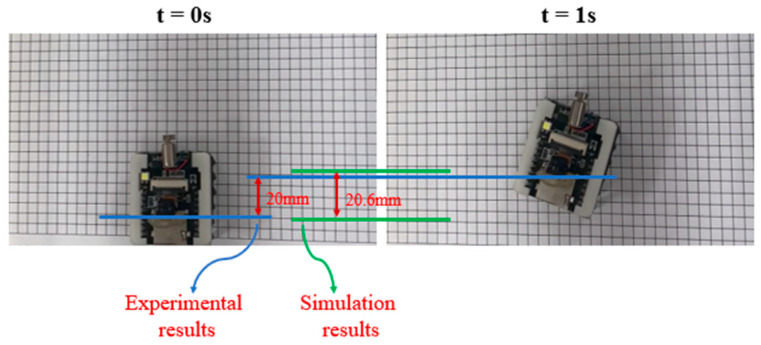
Deviation of the microrobot movement on a grid composed of squares with 5 mm sides, during a 1 s rotation of the motor, from the simulation results during the same period.

**Figure 32 micromachines-15-00202-f032:**
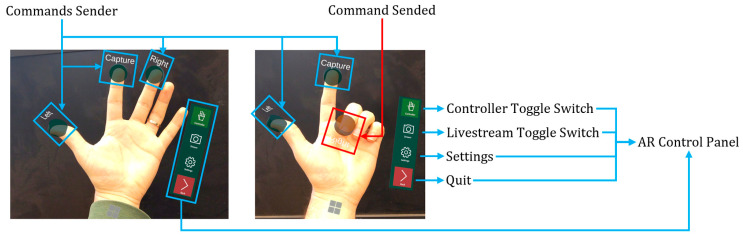
The AR interface for steering the MARSBot is shown in a neutral state on the left, and the AR interface when one button is triggered is shown on the right.

**Figure 33 micromachines-15-00202-f033:**
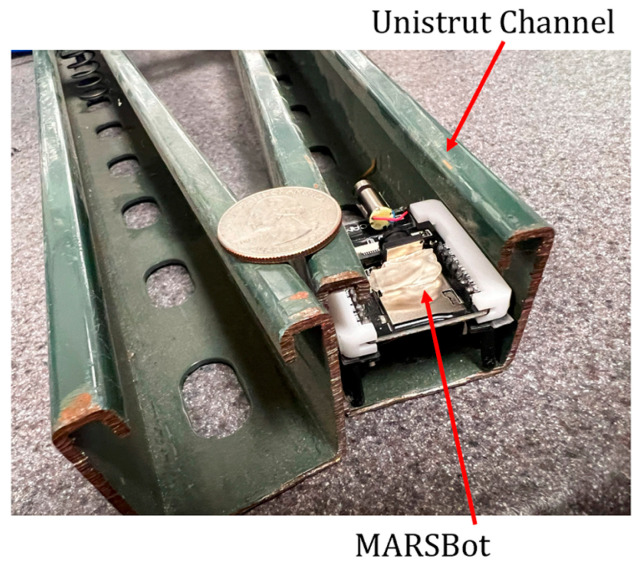
Slotted strut channel inspection by the MARSBot controlled by the PC interface.

**Figure 34 micromachines-15-00202-f034:**
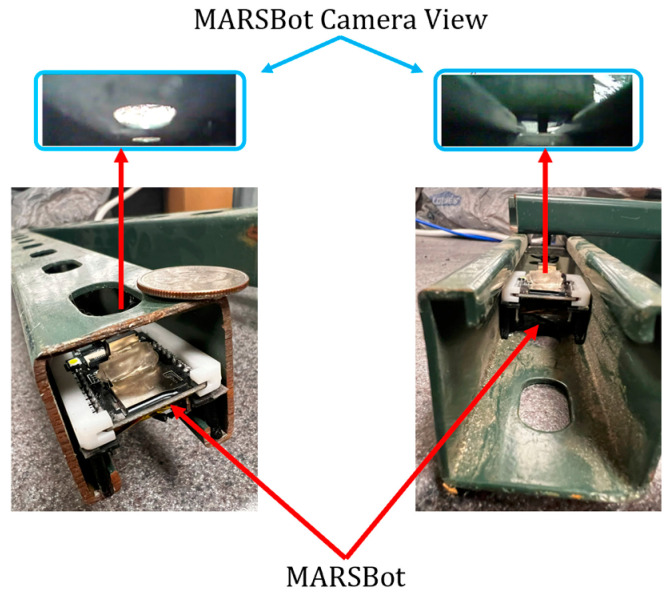
MARSBot visual inspection of the slotted side of the strut channel is displayed on the left, and MARSBot inspection of the connections between strut channels is shown on the right.

**Figure 35 micromachines-15-00202-f035:**
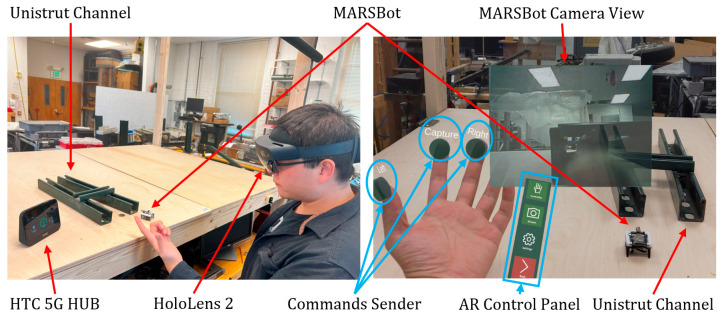
MARSBot visual inspection with AR steering regulated by the operator connected to the same Wi-Fi hotspot is shown on the left, and the AR interface with visual feedback from the MARSBot inspecting Unistrut channels is displayed on the right.

**Table 1 micromachines-15-00202-t001:** Characteristics of the prototypes.

Microrobot	Speed (mm/s)	Dimensions (L × W × H) (mm)	Weight (g)	Durometer
Prototype #1	57.96	45 × 27 × 43	24	64
Prototype #2	-	47 × 33 × 34	22	100
Prototype #3	12.58	45 × 41 × 28	24	44
Prototype #4	68.13	47 × 36 × 30	22	72

**Table 2 micromachines-15-00202-t002:** Force–displacement experiment for the calculation of the stiffness.

Case	Force (N)	Displacement (m)
Experimental Case #1	0.66708	0.000135
Experimental Case #2	1.11834	0.000528
Experimental Case #3	1.53036	0.002284

**Table 3 micromachines-15-00202-t003:** Stiffness calculation based on the manufacturer’s data.

Case	Young’s Modulus (Pa)	Cross-Sectional Area (m^2^)	Length (m)	Stiffness (N/m)
Theoretical	1.8 M	7.06858 × 10^−6^	0.0145	1579

## Data Availability

Data are contained within the article and Appendix A.

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
