# Peer review of "MARSBot: A Bristle-Bot Microrobot with Augmented Reality Steering Control for Wireless Structural Health Monitoring"

_micromachines, 2024, doi:10.3390/mi15020202_

Round 1

Reviewer 1 Report

Comments and Suggestions for Authors

The authors introduce a bristle-bot microrobot in this paper, which utilizes bristlebot locomotion and incorporates a novel centrifugal yaw steering control scheme. The study successfully demonstrates the effectiveness of the robot's motion through a combination of theoretical analysis and experimental verification. The manuscript exhibits a well-structured framework, and the experimental design is deemed reasonable. However, there are certain aspects that necessitate further elucidation by the authors.

1.         As stated in the introduction, there has been significant progress in the development of microrobots in recent years. However, the existing literature lacks a comprehensive and coherent summary of previous studies on the driving and control of microrobots, as well as a thorough comparison with the present study. It is therefore hoped that the following papers focusing on the driving and control of microrobots can provide valuable insights to the authors. To enhance the comprehensiveness of the article, it would be highly beneficial to conduct a comparative analysis the bristle-bot microrobot and other microrobots that do not possess an onboard power supply. This analysis will significantly contribute to improving the overall integrity of the study.

[1] Z. Hao et al., "Controlling Collision-Induced Aggregations in a Swarm of Micro Bristle Robots," in IEEE Transactions on Robotics, vol. 39, no. 1, pp. 590-604, Feb. 2023, doi: 10.1109/TRO.2022.3189846.

[2] S. Zhong et al., "Spatial Constraint-Based Navigation and Emergency Replanning Adaptive Control for Magnetic Helical Microrobots in Dynamic Environments," in IEEE Transactions on Automation Science and Engineering, Early Access, doi: 10.1109/TASE.2023.3339637.

[3] T. Xu et al., "Multimodal Locomotion Control of Needle-Like Microrobots Assembled by Ferromagnetic Nanoparticles," in IEEE/ASME Transactions on Mechatronics, vol. 27, no. 6, pp. 4327-4338, Dec. 2022, doi: 10.1109/TMECH.2022.3155806.

[4] T. Xu, C. Huang, Z. Lai and X. Wu, "Independent Control Strategy of Multiple Magnetic Flexible Millirobots for Position Control and Path Following," in IEEE Transactions on Robotics, vol. 38, no. 5, pp. 2875-2887, Oct. 2022, doi: 10.1109/TRO.2022.3157147.

[5] Z. Zheng et al., "Programmable aniso-electrodeposited modular hydrogel microrobots," Sci. Adv. 8, eade6135(2022). DOI:10.1126/sciadv.ade6135.

[6] Z. Hao, D. Kim, A. R. Mohazab and A. Ansari, "Maneuver at Micro Scale: Steering by Actuation Frequency Control in Micro Bristle Robots," 2020 IEEE International Conference on Robotics and Automation (ICRA), Paris, France, 2020, pp. 10299-10304, doi: 10.1109/ICRA40945.2020.9196694.

2.         Several of the illustrations in the manuscript are deficient in essential annotations and explanations. It is recommended that the author enhance the image annotations by including information such as scale, nomenclature for each component, and other relevant details to comprehensively convey the conveyed information. Notably, Figure 9, 10, 12, 14, 15, 16, 31, 32, 33, among others, require improvements in this regard.

3.         Table 1 provides parameter information for various types of microrobots. I am interested to know whether the authors have investigated the correlation between body size, hardness, and application capability. It appears that as the robot size decreases, its functionality may be compromised, and I am curious about how the authors address this issue.

4.         Figure 13 exhibits a commendable visual processing outcome; however, the manuscript lacks a comprehensive description of the methodology employed in this particular section. Could you kindly provide a detailed explanation?

5.         Some parts of the manuscript contain noticeable grammatical errors, such as mixed case. It is recommended that the authors thoroughly review and revise the manuscript for accuracy and clarity.

6.         In the manuscript, the authors utilize human commands as the upper control signal for the robot. It is important to consider the underlying control system of the robot and evaluate its accuracy. I would like to know if the authors have taken this into account and if any measures have been implemented to ensure precise control of the robot.

Comments on the Quality of English Language

1.         Some parts of the manuscript contain noticeable grammatical errors, such as mixed case. It is recommended that the authors thoroughly review and revise the manuscript for accuracy and clarity.

Reviewer 2 Report

Comments and Suggestions for Authors

The manuscript titled “MARSBot: a Bristle-bot Microrobot with Augmented Reality Steering Control for Wireless Structural Health Monitoring” is an innovative work that seeks to use robotics and augmented reality to answer an existing problem in structural health monitoring. This work is in the prototype and experimentation phase, with significant potential for the surveying industry. This work is an interesting read, but attempts to compare this technology with existing state-of-the-art is glaringly missing in this work. Therefore, I propose that the authors address the following concerns before reconsideration for publication.

Comments / suggested edits:

1.       Section 2.1 makes multiple assumptions when deriving the equation of motion for the MARSBot. The authors should clearly state any deviation between what was derived and programmed with the eventual outcome. This could mean including the theoretical value in the respective plots.

2.       The overall graphic processing of the figures needs improvement. Currently, all the figures are of different sizes and text sizes due to inadequate scaling. Significant graphic improvements are required. It would also improve the paper if the figure captions were more self-contained. The current figure captions are simple 1-liners. The authors should consider a sentence or two saying what is the main message of each figure.

3.       The authors submitted two videos demonstrating the use of their innovation. There appears to be a lag between the actual input (user moving one’s fingers), the AR responding to the input, and the microrobot responding to the input. As part of the discussion, the authors should quantify the lag.

4.       To further enhance the impact of your work, I would like to suggest considering a comparison with the current state-of-the-art in the field of SHM (size of microrobot, weight, range, etc.). Providing this context would help readers better understand the significance of your contributions and position your research within the broader landscape of the field. I recommend dedicating a section, perhaps in the discussion, to a comparative analysis where you can discuss how your findings compare to existing benchmarks or widely accepted models.

Minor edits:

1.       There are numerous elementary equations present throughout section 2.1. While I appreciate the mathematical detail, I believe some of these equations are fundamentally overwhelming and might be more suitable for the supplementary information. This adjustment would not only improve the flow of the paper but also enhance its accessibility to a broader audience.

2.       The identification of the research gap and the placement of your prototype within the existing R&D landscape could benefit from further clarification. To enhance the overall clarity of your contribution, I suggest explicitly defining the research gap you aim to address. Additionally, providing more context on how your prototype fits into the crowded R&D landscape would greatly benefit readers in understanding the uniqueness and relevance of your work.

Round 2

Reviewer 2 Report

Comments and Suggestions for Authors

The authors have satisfactorily addressed my queries and edited the manuscript accordingly.